# LC/MS-Based Untargeted Metabolomics Analysis in Women with Morbid Obesity and Associated Type 2 Diabetes Mellitus

**DOI:** 10.3390/ijms24097761

**Published:** 2023-04-24

**Authors:** Teresa Auguet, Laia Bertran, Jordi Capellades, Sonia Abelló, Carmen Aguilar, Fàtima Sabench, Daniel del Castillo, Xavier Correig, Oscar Yanes, Cristóbal Richart

**Affiliations:** 1Grup de Recerca GEMMAIR (AGAUR)-Medicina Aplicada, Departament de Medicina i Cirurgia, Universitat Rovira i Virgili (URV), IISPV, 43005 Tarragona, Spain; mariateresa.auguet@urv.cat (T.A.); laia.bertran@urv.cat (L.B.); carmenisabel.aguilar@urv.cat (C.A.); fatima.sabench@urv.cat (F.S.); danieldel.castillo@urv.cat (D.d.C.); 2Department of Electronic Engineering, Universitat Rovira i Virgili (URV), IISPV, 43007 Tarragona, Spain; jordi.capellades@iispv.cat (J.C.); xavier.correig@urv.cat (X.C.); oscar.yanes@urv.cat (O.Y.); 3Servei de Recursos Científics i Tècnics, Universitat Rovira i Virgili (URV), 43007 Tarragona, Spain; sonia.abello@urv.cat; 4Unitat de Cirurgia, Facultad de Medicina i Ciències de la Salut, Hospital Universitari Sant Joan de Reus, Universitat Rovira i Virgili (URV), IISPV, 43204 Reus, Spain; 5CIBER de Diabetes y Enfermedades Metabólicas Asociadas, Instituto de Salud Carlos III, 43204 Madrid, Spain

**Keywords:** obesity, diabetes, metabolites, metabolomics

## Abstract

Obesity is a chronic and complex disease, with an increasing incidence worldwide that is associated with metabolic disorders such as type 2 diabetes mellitus (T2DM). Thus, it is important to determine the differences between metabolically healthy obese individuals and those with metabolic disorders. The aim of this study was to perform an untargeted metabolomics assay in women with morbid obesity (MO) compared to a normal weight group, and to differentiate the metabolome of these women with MO who present with T2DM. We carried out a liquid chromatography-mass spectrometry-based untargeted metabolomics assay using serum samples of 209 Caucasian women: 73 with normal weight and 136 with MO, of which 71 had T2DM. First, we found increased levels of choline and acylglycerols and lower levels of bile acids, steroids, ceramides, glycosphingolipids, lysophosphatidylcholines, and lysophosphatidylethanolamines in MO women than in the control group. Then, in MO women with T2DM, we found increased levels of glutamate, propionyl-carnitine, bile acids, ceramides, lysophosphatidylcholine 14:0, phosphatidylinositols and phosphoethanolamines, and lower levels of Phe-Ile/Leu. Thus, we found metabolites with opposite trends of concentration in the two metabolomic analyses. These metabolites could be considered possible new factors of study in the pathogenesis of MO and associated T2DM in women.

## 1. Introduction

Obesity is a chronic and complex disease that can induce negative effects on health [1,2]. The incidence of obesity has increased considerably worldwide in recent decades [3]. Sustained high-carbohydrate or saturated fat diets tend to induce obesity but also insulin resistance and, consequently, disorders such as type 2 diabetes mellitus (T2DM) [4]. T2DM is a growing global health problem that is closely linked to obesity epidemics [5]. In this sense, it appears that the current society is burdened by a pandemic of chronic diseases in which metabolic dysregulation plays a key role in pathogenesis and progression, including obesity, T2DM, and cardiovascular diseases [6]. While there are patients with obesity who suffer from metabolic alterations that entail a cardiovascular risk, others do not present metabolic alterations [7]. Therefore, it is important to determine the differential metabolites between metabolically healthy obese patients, and those with metabolic disorders to find predictive biomarkers [8,9,10].

As such, metabolomics is a unique tool to measure low-weight molecules found in human body fluids, such as blood or urine, using an analytical chemistry method based on a gas or liquid chromatography separation technology coupled to a mass spectrometer [11]. Metabolomics is able to identify characteristic metabolites or metabolic patterns from a specific condition, and thus aids in understanding the precise pathophysiologic mechanisms involved in early metabolic disorders [6]. In this regard, nontargeted metabolomics, which consists of the unbiased comparison of the metabolome between different biological circumstances, is a suitable strategy for identifying and quantifying metabolites to find potential biomarkers [6].

Previous metabolomics studies on obesity reported differentially concentrated metabolites in different metabolic conditions, such as branched-chain amino acids (BCAAs), non-esterified fatty acids, organic acids, acylcarnitines, and phospholipids, which were investigated as potential biomarkers for obesity [12,13]. In addition, altered levels of lysoglycerophospholipids were shown to be associated with morbid obesity (MO) [14]. On the other hand, the levels of alanine, glutamate, and palmitic acid were significantly different in T2DM and pre-diabetes [9].

However, many of these studies have small and heterogeneous cohorts with respect to age, sex, and type of obesity. Furthermore, it is important to note that most of these investigations have been performed by targeted metabolomics, and thus mostly measured a recurrent small set of metabolites [12,13].

For this reason, the aim of this study was to carry out a liquid chromatography-mass spectrometry (LC/MS)-based untargeted metabolomics in serum samples of women with MO compared to normal weight healthy women to define the precise metabolic profiling of those subjects, and differentiate the metabolome between MO women who are metabolically healthy and those who present with associated T2DM.

## 2. Results

### 2.1. General Classification of Participants

We performed an LC/MS-based untargeted metabolomics study in serum samples from 209 women. This cohort was composed of 73 healthy women with normal weight, who were categorized as the control group. The study group had 136 women with MO, 65 of whom were metabolically healthy and 71 with associated T2DM.

First, the clinical and anthropometrical parameters of the studied cohort were evaluated, as shown in Table 1. Subjects were comparable in terms of age and sex (all were women). MO subjects presented significantly higher BMI, waist–hip ratio, glucose, HbA1c, insulin and triglyceride levels, and lower levels of HDL-C than the normal weight group. However, MO patients also presented lower levels of total cholesterol and non-significant differences regarding LDL-C, which can be due to most of MO subjects were treated with lipid-lowering agents.

### 2.2. Metabolic Profile in Morbidly Obese Women in Comparison with Healthy Normal Weight Women

When we evaluated the metabolic profile in MO participants compared to the normal weight group, we found significantly differently concentrated metabolites (Table 2). MO subjects presented significantly higher levels of carbohydrate metabolites except for iditol. In addition, we found higher levels of choline and some amino acids, and lower levels of other amino acids compared to the control group. The same was found with peptides, with higher levels of the tripeptide glycine-glycine-valine (Gly-Gly-Val) in MO subjects; however, lower levels of dipeptides in comparison with normal weight subjects.

Regarding lipids and derivatives, MO women presented increased levels of monoacylglycerols, diacylglycerols, and triglycerides compared to normal weight subjects (Table 3A). We found lower levels of bile acids, steroids, ceramides, lysophosphatidylcholines, and lysophosphatidylethanolamines (Table 3B).

In addition, families of metabolites with some species increased, and others decreased, such as acylcarnitines, sphingomyelins, fatty acids, phosphatidylcholines, phosphatidylinositols, and phosphoethanolamines.

We performed a principal component analysis (PCA) to depict the multifactorial characteristics of the MO phenotype compared to women with normal weight. In this sense, there are considerable dissimilarities in the metabolite abundances between individuals of the MO group, while the normal weight group remains more consistent in the model (Figure 1). Such dissimilarities within the MO group may be a result of the diverse genetic background and lifestyle differences of the patients, even if the variables used are significantly abundant within the normal weight and MO groups.

### 2.3. Metabolic Profile in Morbidly Obese Women with Type 2 Diabetes Mellitus in Comparison with Those Metabolically Healthy

Concerning metabolomics in MO patients presenting with T2DM, we found significantly higher levels of carbohydrate metabolites such as anhydrohexose, glucose, and sorbitol than in MO patients without T2DM, while we found lower levels of iditol (Table 4). In addition, we reported higher levels of glutamate and lower levels of the dipeptide Phe-Ile/Leu in subjects with T2DM associated with MO. Concerning lipids and derivatives, T2DM women with MO presented higher levels of propionyl-carnitine, bile acids, ceramides, LPC 14:0, PI, PE and PC, except for PC 32:1 and PC 34:4, which were found to be decreased.

Later, we performed another PCA to depict the multifactorial characteristics of T2DM associated with the MO phenotype, compared to metabolically healthy morbid obesity and normal weight. In this regard, the PCA model allows us to differentiate between the group of patients with normal weight, from the metabolically healthy morbid obese women and those with T2DM, but not between metabolically healthy obese women and diabetic women (Figure 2).

### 2.4. Comparison between Women with Morbid Obesity versus Normal Weight Analysis and Type 2 Diabetes Mellitus Morbid Obese Women versus Those Metabolically Healthy Analysis

Comparing what we found in the MO metabolome analysis with what we found in the T2DM associated with obesity analysis, we reported similar tendencies regarding carbohydrates and peptides, since we found increased levels of anhydrohexose in MO and in T2DM subjects, and decreased levels of iditol and dipeptide Phe-Leu/Ile in the MO and T2DM associated with MO groups (Table 2 and Table 4). Regarding lipids and derivatives, we found a contrary tendency in terms of bile acids, ceramides, lysophosphatidylcholines, phosphatidylcholines PC 30:0, PC 32:0, PC 32:2, phosphatidylinositols PI 36:3, and phosphoethanolamines PE 34:3 and PE 40:5, given that we found decreased levels of these metabolites in MO subjects; however, we found them increased in the T2DM associated with MO group (Figure 3).

We found that PC 32:1 and PC 34:3 levels decreased in both conditions. Our comparison showed that PE 34:2, PE 38:5, PE 38:6, and PE 40:6 levels were higher in both groups than in their controls. In this sense, we can observe that most of the metabolites in the MO analysis were decreased compared to the normal weight group, while most of the metabolites increased in the MO with T2DM women group in comparison with metabolically healthy MO women.

## 3. Discussion

The novelty of this work is that we performed an LC/MS-based untargeted metabolomics study using serum samples from a well-characterized cohort of women with T2DM associated with MO. Here, we first compared the metabolomics profile of women with MO in comparison with healthy normal weight women. In this regard, we found increased levels of choline, monoacylglycerols, diacylglycerols, and triglycerides in MO women. We observed lower levels of bile acids, steroids, ceramides, glycosphingolipids, lysophosphatidylcholines, and lysophosphatidylethanolamines in MO women.

Concerning this analysis, we classified the metabolites that we found into three groups: (1) carbohydrates, (2) amino acids, peptides and choline, and (3) lipids and derivatives. Regarding carbohydrates, our MO women presented with higher levels of anhydrohexose and mannose, and lower levels of iditol than the normal weight group. In this way, Moore, et al., and Xu, et al., found a positive correlation between mannose levels and BMI and glucose levels in the blood [15,16].

Abnormal peptide and amino acid metabolism have been associated with impaired protein processing and accumulation of toxic intermediates that promote insulin resistance, and consequently obesity [17,18]. Regarding the amino acids, peptides and choline that we found in the metabolomic study, MO subjects presented with higher levels of acetyl-alanine and tyrosine, and lower levels of N-acetyl-serine and phenyl-glutamine. In this sense, previous metabolomic analyses reported higher levels of alanine and tyrosine in obese subjects [15,19,20,21]. In addition, we found higher levels of choline in MO women, which agrees with our previous study [22] and with Gogna, et al., who also reported increased levels of choline in high BMI subjects [23]. This fact is reinforced by an animal model study that reported choline deficiency as an attenuator of body weight gain in obese mice [24].

In addition, we observed that the oligopeptide Gly-Gly-Val increased its levels, while the dipeptides Phe-Phe, Phe-Trp, Ile-Ile/Leu-Leu and Phe-Ile/Leu decreased in MO subjects compared to the normal weight group. Contrary to our findings, BCAAs (Val, Ile and Leu) and Phe were increased in obesity, while Gly was decreased [25]. Regarding Trp levels, it has been reported that Trp metabolism is altered in obesity, with subjects exhibiting low levels of this amino acid [26]. These mentioned discrepancies can be understood given the variability of the metabolomic technique or the study cohort, since it can be composed of subjects of one or both sexes, and also affected by the ethnicity or type of obesity that previous articles used to evaluate subjects with obesity. In our study, we recruited only MO women.

In terms of lipids and their derivatives, some authors’ findings were consistent with ours, reporting high levels of fatty acids in obesity, specifically concerning oleic acid levels [27,28,29]. In the case of dodecanoic acid, also known as lauric acid, which is a medium-chain fatty acid present in coconut oil that was reported to have beneficial effects in obese subjects [30], we found it to be decreased in MO women compared to the control group, which makes sense with the literature.

The characteristic free fatty acid oversupply characteristic of obesity tends to cause an incomplete induction of fatty acid oxidation pathway, triggering the accumulation of intermediates such as acylcarnitines and the deficiency of free carnitine [31]. In this sense, previous metabolomic reports found higher levels of acylcarnitines in obese subjects [6,13,15,32,33,34,35]. Specifically, palmitoyl-carnitine, which is a marker of impaired fatty acid oxidation [36], and oleoyl carnitine, which were found to be increased in obese subjects [37], were increased in this study. Zhang, et al., reported decreased levels of butyryl-carnitine in obese mouse models [38], which agrees with our results. Kang, et al., found a negative association between octanoyl-carnitine levels and visceral fat area [39], which matches our finding regarding acyl-carnitine.

In a situation of fatty acid oxidation impairment, non-oxidized fatty acids may be converted into other lipids, such as ceramides or acylglycerols, which can negatively affect insulin signalling [12]. In this regard, we found increased levels of monoacylglycerols, diacylglycerols, and triacylglycerols in MO subjects, which agrees with a recent article by Yin, et al., who reported higher levels of diacylglycerols and triacylglycerols in obesity [40]. In the case of ceramides, we found that they decreased in our MO women in comparison with the control group; however, an increase in ceramides was previously reported in obesity or metabolic diseases [41,42]. In any case, Pikó, et al., found an inverse association between HexCer levels and BMI [43], reinforcing our results.

Concerning bile acids, which are involved in the intestinal absorption of fats and related to insulin resistance and obesity [44], we found lower levels of deoxycholic acid, glycoursodeoxycholic acid, and glycocholic acid in MO women than in the normal weight group. In our own previous study, we reported lower levels of deoxycholic acid in obese subjects, according to Prinz, et al., [45]; however, we did not find significant differences in glycocholic acid or glycoursodeoxycholic acid levels [22]. In addition, here we found lower levels of steroids such as pregnanolone sulfate, androsterone glucuronide, and cholesterol 3-sulfate in MO women compared to the normal weight group, which seems similar to the literature, since reduced levels of androsterone glucuronide and cholesterol 3-sulfate were associated with obesity [46,47]. With regard to sphingomyelins, which have been related to insulin signalling inhibition and induction of inflammation [48], we found that some species increased, such as SM 36:1 and SM 36:2, while we found decreased levels of SM 34:1, which compared to the literature, a general increase in sphingomyelins in obesity has been reported [43].

In relation to the complex phospholipids found in cell membranes [49], in this study, we found lower levels of lysophosphatidylcholines and lysophosphatidylethanolamines in MO women than in the normal weight group. Considering previous publications, Kim, et al., reported low levels of LPC 18:1 in obesity [32], while other authors observed low levels of LPC 18:1, LPC 18:2, and LPC 20:4 in obesity [33,50,51], which coincides with our results. Nonetheless, Kim, et al., also reported higher levels of LPC 14:0, LPC 16:0, and LPC 18:0, contrary to our findings [32]. In the case of lysophosphatidylethanolamines, our findings agree with previous publications [27,32].

Regarding phosphatidylcholines, we found higher levels of PC 31:2, PC 31:3, PC 36:4, PC 38:4, and PC 40:6 in MO women. Pietiläinen, et al., coincides with our findings of PC 36:4 and PC 38:4; however, discerns concerning PC 40:6 [52]. In the case of phosphatidylinositols and phosphoethanolamines, we found that some species increased while others decreased; other authors stated that phosphoethanolamines were decreased in obesity [27,32], and phosphatidylinositols seemed to play an anti-obesity role in mouse models [53].

Then, we evaluated the metabolome of MO women with T2DM in comparison with that of metabolically healthy MO women. Accordingly, we found increased levels of glutamate, propionyl-carnitine, bile acids, ceramides, lysophosphatidylcholine LPC 14:0, phosphatidylinositols, and phosphoethanolamines in T2DM associated with MO subjects. In addition, we found lower levels of the dipeptide Phe-Ile/Leu in women with T2DM with MO.

Regarding carbohydrates, we found higher levels of anhydrohexose, glucose and sorbitol, and lower levels of iditol compared to metabolically healthy MO women. In this regard, given that glucose is the main marker of T2DM [54], our results agree with several previous metabolomic reports with similar cohorts of obese subjects with T2DM [23,27,55]. Elevated levels of sorbitol in serum were previously found in two different cohorts of patients (men and women) with obesity and associated T2DM [23,56].

With respect to the amino acids, peptides, and choline group, we found higher levels of glutamate and lower levels of the dipeptide Phe-Ile/Leu in subjects with T2DM. In this regard, Lee, et al., stated that high glutamate levels are associated with the development of T2DM [57]. However, Libert, et al., found decreased levels of glutamate in T2DM [21]. This same report declared that Phe and Leu levels were increased in diabetic subjects, which contradicts our results regarding T2DM metabolomic analysis. These contradictions can be explained, as we already said, by the variability of the metabolomic technique or the characteristics of the study cohort.

Concerning the last group composed of lipids and derivatives, we found increased levels of propionyl-carnitine in MO women with T2DM. In this way, Adams, et al., confirmed our results in their study of obese and diabetic American–African women [58]. In the case of bile acids, we found higher levels of glycoursodeoxycholic acid and glycocholic acid in MO women with T2DM. Bile acids are increased in T2DM since they are implicated in the regulation of glucose metabolism and in endoplasmic reticulum stress, leading to insulin resistance [59]. However, it was previously reported that there were non-significant differences regarding glycocholic acid or glycoursodeoxycholic acid between diabetic and nondiabetic subjects [60], and glycoursodeoxycholic acid levels were found to be decreased in subjects with high blood glucose levels [61]. In this regard, these discrepancies can be explained because we evaluated T2DM presence in a cohort of women with MO. In terms of ceramides, which we found to be increased in T2DM associated with obesity patients, it agrees with the literature, since metabolically altered obese subjects used to present higher levels of these lipids [41,42]. In the case of the lysophosphatidylcholine LPC 14:0, which was found to be increased in the diabetic subjects of the current study, Liu, et al., in a human model, and Kim, et al., in an animal model, stated decreased levels of lysophosphatidylcholines in T2DM associated with obesity [20,28], given that this lipid derivative seems to induce the cellular uptake of glucose, reducing blood glucose levels [62]. Phosphatidylcholines, which presented in some species as increased and others as decreased in our T2DM subjects, are usually increased in diabetic obese subjects [20,52,55,63]. Considering phosphatidylinositols and phosphoethanolamines, we reported some increased species and other decreased species in T2DM associated with MO women, which is difficult to compare with the literature since the authors did not describe these metabolites. However, phosphoethanolamines were found to be positively associated with changes in insulin resistance and β-cell function and inversely associated with changes in insulin sensitivity [64], while phosphatidylinositols used to be decreased in T2DM due to the impairment in the regulation of the phosphatidylinositol 3-kinase system [65].

In summary, we want to highlight the most relevant aspects of this article, which is the significant decrease in bile acids, ceramides, lysophosphatidylcholines, phosphatidylinositol, and most phosphatidylcholines levels in MO women compared to the control group. In contrast, in women with T2DM associated with MO, we found a relevant increase in the levels of these metabolites. We observed an increase in the levels of most of the phosphoethanolamines in both analyses, except for PE 34:3 and PE 40:5, which showed opposite trends in both analyses. On the other hand, we observed decreased levels in both evaluations concerning PC 32:1 and PC 34:3. In this sense, these metabolites with opposite trends of concentration between women with MO, compared to the control group and women with T2DM associated with MO, compared to metabolically healthy MO, allow us to discern the metabolomic profile of these patients with T2DM associated with obesity from those without metabolic alterations. However, although bile acids have an insulin-sensitizing effect and facilitate intestinal fat absorption, being involved in carbohydrate and lipid metabolism, which contribute to obesity and T2DM pathogenesis, their precise role remains unclear [44,66]. In the same line, certain ceramides and phospholipids such as phosphatidylcholines, lysophosphatidylcholines, phosphoethanolamines, and phosphatidylinositol have a lipotoxic role associated with the state of chronic low-grade inflammation of obesity and T2DM [67,68]. However, some species have demonstrated to present a potential anti-inflammatory role in a deregulated metabolism condition [69]. For this reason, and the fact that there are few related works of literature and lack of homogeneity in the studied cohorts, it makes it difficult to discuss the specific role of these lipidic species in obesity and T2DM pathologies. More studies are needed to define the specific implication of each metabolite.

Thus, although we have conducted a study in a homogeneous cohort of morbidly obese women, it is difficult to compare these results with the literature due to the differences between the studied cohorts, since they can generate a bias. Moreover, this study has a small cohort of subjects, and we did not evaluate at the molecular level the genes that can be involved in the susceptibility to develop MO, insulin resistance and T2DM, and diseases induced by the interaction of genetic, epigenetic, and environmental factors [70,71,72], so it is necessary to perform more specific or targeted omics analyses in other well-characterized cohorts or in larger validation cohorts to confirm these findings.

## 4. Materials and Methods

### 4.1. Subjects

In the present work, we carried out an LC/MS-based untargeted metabolomics assay in 209 Caucasian women, 73 healthy women with normal weight (BMI 19–25 kg/m^2^), and 136 women with MO (BMI ≥ 40 kg/m^2^) who were scheduled to undergo laparoscopic bariatric surgery. Our cohort was made up of only women because we wanted to study a homogenous cohort to avoid the interference of several confounding factors. It is well known that men and women differ substantially regarding body composition, energy imbalance, and hormones. Moreover, several studies have shown sex-specific differences in lipid and glucose metabolism [73,74].

The study was approved by the institutional review board (Institut Investigació Sanitària Pere Virgili (IISPV) CEIm; 23c/2015), and all participants gave written informed consent. Patients who had an acute illness, acute or chronic inflammatory or infective diseases, or end-stage malignant disease were excluded from this study. Menopausal women and women receiving contraceptive treatment were also excluded.

### 4.2. Subclassification of the Cohort

Of the 136 patients with MO, 71 had been diagnosed with T2DM meeting the diagnostic criteria of the American Diabetes Association (ADA) [75]. On the other hand, the other 65 MO women were metabolically healthy, since we initially excluded the patients with metabolic syndrome or pre-diabetes.

### 4.3. Anthropometrical and Biochemical Analysis

Anthropometrical evaluation included measurement of weight, height, waist–hip ratio, and BMI calculation. Blood extraction was performed by specialized nurses through a BD Vacutainer^®^ system (Franklin Lakes, NJ, USA) after overnight fasting and just before bariatric surgery. Venous blood samples were obtained in ethylenediaminetetraacetic acid tubes, which were separated into plasma and serum aliquots by centrifugation (3500 rpm, 4 °C, 15 min), and then stored at −80 °C until processing. Biochemical studies included glucose, insulin, glycated haemoglobin A1c (HbA1c), total cholesterol, high-density lipoprotein-cholesterol (HDL-C), low-density lipoprotein-cholesterol (LDL-C), and triglyceride levels, which were carried out using a conventional automated analyzer and measured after overnight fasting.

### 4.4. LC/MS Methods

#### 4.4.1. Metabolite Extraction

Sample metabolites were extracted by mixing 25 μL of plasma and 300 μL of cold acetonitrile:methanol:water (5:4:1/*v*:*v*:*v*) and then shaken with a vortex. Samples were kept on ice for 30 min and later centrifuged at 12,000 rpm for 10 min at 4 °C. The supernatants were placed in vials suitable for LC/MS analysis.

#### 4.4.2. LC/MS Settings

Samples were analyzed on a Thermo Scientific™ Orbitrap IDX Tribrid mass spectrometer (Thermo Scientific, Waltham, MA, USA) with a HESI interface, in line with the Vanquish UHPLC Liquid Chromatograph. An Acquity UPLC BEH HILIC column (2.1 × 150 mm × 1.7 µm) provided by Waters was used. ACN (A) and ammonium acetate 50 mM (B) were used for gradient elution. The column temperature and flow were set to 25 °C and 0.4 mL/min, respectively. An injection volume of 5 μL was used. The optimized gradient was programmed as follows: starting with 5% B, the gradient remained isocratic for 2 min, then increased within 4 min to 50% B, and stayed constant for 1 min. For re-equilibration, the gradient was changed to starting conditions within 0.2 min and held constant for 3.2 min. The total run time was 10.5 min.

For MS detection, heated electrospray ionization settings were set in positive and negative ionization modes as follows: source voltage, 3.5 kV (positive), 2.8 kV (negative); ion transfer tube, 300 °C; vaporizer temperature, 300 °C; sheath gas (N2) flow rate, 50 a.u.; auxiliary gas (N2) flow rate, 10 a.u.; sweep gas (N2) flow rate, 1 a.u.; s-lens rf level, 60%; SCAN-mode; resolution, 120,000 (at *m*/*z* 200); AGC target, 50%; and maximum injection time, 200 ms.

MS/MS acquisition was performed at a resolution of 15,000, and the normalized collision energy of the HCD cell was fixed at 35%. This collision energy was tested to obtain appropriate precursor ion and product intensities. The quadrupole isolation window was 1 *m*/*z*. The maximum injection time of the C-trap was customized for each inclusion list.

Xcalibur 4.4 software (Thermo Scientific, Waltham, MA, USA) was used for LC/MS instrument control and data processing. Quality control (QC) samples consisting of pooled samples from each condition were injected at the beginning and periodically throughout the workflow.

#### 4.4.3. Metabolite Identification by MS/MS

Identification was performed by HERMES using two strategies: cosine spectral matching using an in-house DB, containing MS/MS spectra from MassBankEU, MoNA, HMBD, Riken and NIST14 databases, and using MassFrontier version 8.0 SR1 (Thermo Scientific, Waltham, MA, USA) matching against the mzCloud database. Spectral hits with high similarity scores (>0.8) were manually revised to assess correct metabolite identifications.

### 4.5. Statistical Analysis

All the values reported are expressed as median and interquartile range or mean and standard deviation in accordance with the distribution of the values of the variables. Differences between groups were calculated using the nonparametric Mann–Whitney test. *p* values < 0.05 were considered to be statistically significant.

LC/MS data were processed using the HERMES R package [76] for MS1 profiling (using a database containing 22,314 unique molecular formulas from ChEBI and HMDB), quantification, and inclusion list generation for metabolite identification. Only SOI ions quantified in >80% of the samples were statistically tested for significant differences across the experimental groups using one-way ANOVA. Statistical results were then adjusted using the false discovery rate (FDR) *p*-value correction method. Data from both analyses of significant metabolites concentration expressed in the log, mean, and standard deviation are given in the Appendix A.

## 5. Conclusions

In this LC/MS-based untargeted metabolomics study, by means of a well-characterized cohort of women with MO, we found increased levels of choline and different metabolites from the class of lipids and derivatives: monoacylglycerols, diacylglycerols, and triacylglycerols. Additionally, we found decreased metabolite levels of this class compared to the control group: bile acids, steroids, ceramides, glycosphingolipids, lysophosphatidylcholines, and lysophosphatidylethanolamines. In contrast, we found increased levels of glutamate, propionyl-carnitine, bile acids, ceramides, lysophosphatidylcholine LPC 14:0, phosphatidylinositols, and phosphoethanolamines, and lower levels of the dipeptide Phe-Ile/Leu in MO women with T2DM. In this sense, we found a metabolomic profile in MO women that was distinct from that in MO women with T2DM, presenting opposite trends in most of the significantly concentrated metabolite levels. Therefore, awaiting new investigations, these metabolites could be considered as new factors of study in the pathogenesis of morbid obesity and associated type 2 diabetes mellitus in women.

## Figures and Tables

**Figure 1 ijms-24-07761-f001:**
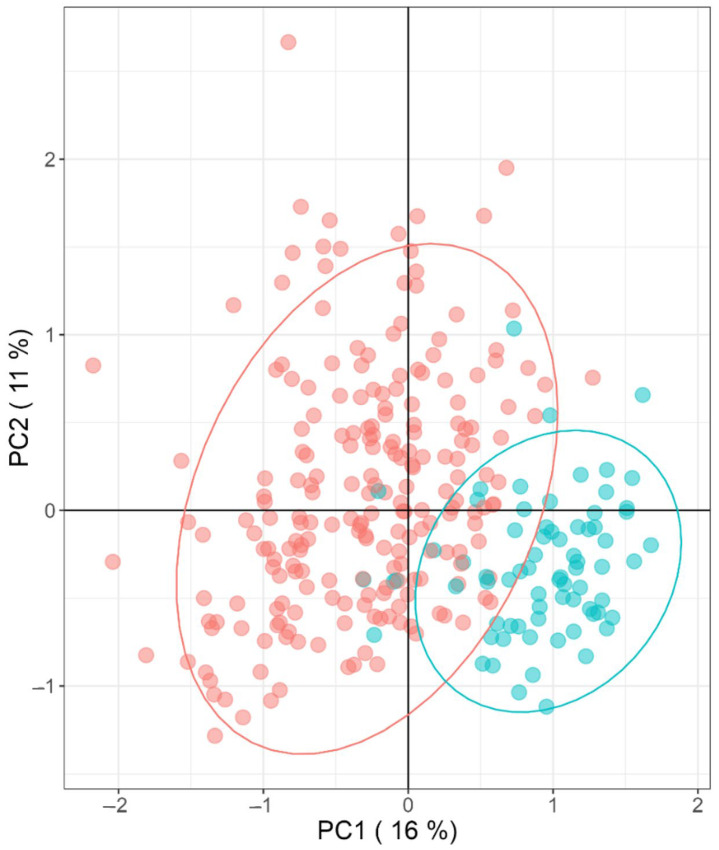
Principal Component Analysis (PCA) of significantly different metabolites between groups (morbidly obese (red) and normal weight (blue)).

**Figure 2 ijms-24-07761-f002:**
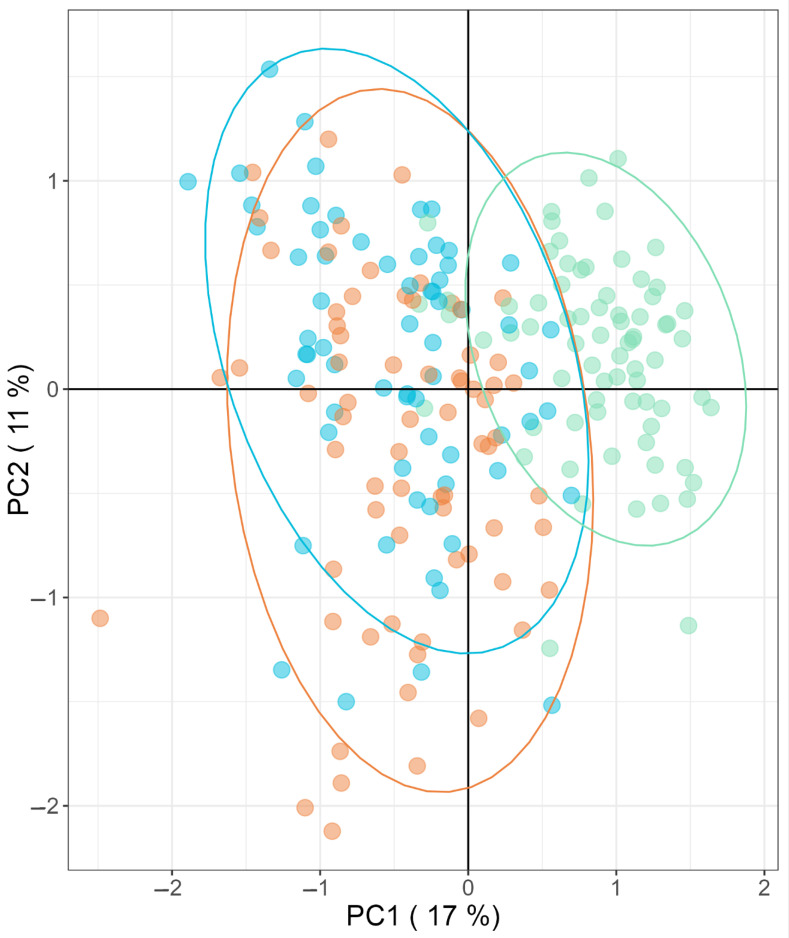
Principal Component Analysis (PCA) of significantly different metabolites between groups (normal weight (green), metabolically healthy morbid obesity (blue), and morbid obesity with associated T2DM (orange)).

**Figure 3 ijms-24-07761-f003:**
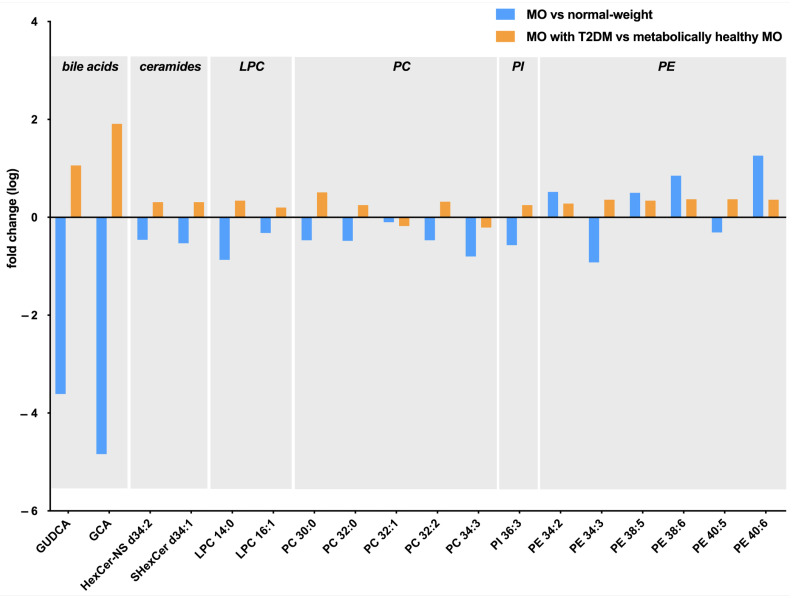
Graphical representation of the metabolites abundance in morbidly obese (MO) vs. normal weight analysis (blue bars), and in the MO with type 2 diabetes mellitus (T2DM) vs. metabolically healthy MO (orange bars). GUDCA, glycoursodeoxycholic acid; GCA, glycocholic acid; HexCer, hexosylceramide; LPC, lysophosphatidylcholines; PC, phosphatidylcholines; PI, phosphatidylinositols; PE phosphoethanolamines.

**Table 1 ijms-24-07761-t001:** Anthropometric and biochemical data of the study group (**A**) Normal weight and morbid obesity comparison; (**B**) Metabolically healthy morbid obesity and morbid obesity with type 2 diabetes mellitus comparison.

(A)					
Variables	Normal Weight (*n* = 73) Median (25th–75th)		Morbid Obesity (*n* = 136) Median (25th–75th)		*p*-Value
Age (years)	40	(36–48)	43	(36–49)	0.497
BMI (kg/m^2^)	22.58	(21.31–23.84)	45.30	(42.23–49.71)	<0.001
Waist-hip (m) ratio	0.79	(0.73–0.85)	0.92	(0.86–0.97)	<0.001
Glucose (mg/dL)	82	(72–91)	124	(114–138)	<0.001
HbA1c (%)	5	(4.75–5.25)	6.1	(5.32–7.80)	<0.001
Insulin (mUI/L)	5.93	(4.54–8.94)	14.1	(8.19–23.48)	<0.001
Cholesterol (mg/dL)	180	(162–205.25)	165.55	(146.80–188.85)	0.005
HDL-C (mg/dL)	65.1	(54.75–73)	39.85	(33.95–46)	<0.001
LDL-C (mg/dL)	105	(86.50–123)	98.50	(83.12–115.75)	0.151
Triglycerides (mg/dL)	63	(50.50–85.50)	129	(99–158)	<0.001
**(B)**					
**Variables**	**Metabolically Healthy** **Morbid Obesity** **(*n* = 65)** **Median (25th–75th)**		**Morbid Obesity with Type** **2 Diabetes Mellitus** **(*n* = 71)** **Median (25th–75th)**		***p*-Value**
Age (years)	46	(40–54)	49	(45–55)	0.063
BMI (kg/m^2^)	44.58	(41.66–48.40)	45.45	(43.48–50.52)	0.105
Waist-hip (m) ratio	0.92	(0.84–0.95)	0.94	(0.88–0.99)	0.053
Glucose (mg/dL)	87	(80.50–94.50)	145.50	(127.75–190.50)	<0.001
HbA1c (%)	5.3	(5–5.65)	7.2	(6.30–8.70)	<0.001
Insulin (mUI/L)	10	(6.64–16.50)	19.33	(13.02–32.48)	<0.001
Cholesterol (mg/dL)	164	(146.37–187.50)	166.85	(147.62–190.40)	0.831
HDL-C (mg/dL)	42	(34.95–53.65)	37.1	(33–44.50)	0.023
LDL-C (mg/dL)	100	(85.15–119.50)	95	(76.40–114)	0.164
Triglycerides (mg/dL)	104	(82.25–132.25)	148	(122–207)	<0.001

Data are expressed as median and interquartile range. Significant differences were considered when *p*-value < 0.05. BMI, body mass index; HbA1c, glycosylated hemoglobin A1c; HDL-C, high-density lipoprotein-cholesterol; LDL-C, low-density lipoprotein-cholesterol.

**Table 2 ijms-24-07761-t002:** Metabolite levels increased or decreased in serum samples of MO patients compared to normal weight subjects.

Group	Class	Increased Metabolite Levels	Decreased Metabolite Levels
Carbohydrates	Sugars	Anhydrohexose, Mannose	Iditol
Amino acids, peptides and choline	Amino acids	Acetyl-alanine, Tyrosine	N-Acetyl-serine, Phenylacetylglutamine
	Peptides	Gly-Gly-Val	Phe-Phe, Phe-Trp, Ile-Ile/Leu-Leu, Phe-Ile/Leu
	Choline	Choline	
Lipids and derivatives	Acylcarnitines	Palmitoyl-carnitine, Oleoyl-carnitine, Acetyl-carnitine, Hydroxybutyryl-carnitine	Butyryl-carnitine, Octanoyl-carnitine, Decanoyl-L-carnitine
	Bile acids		Deoxycholic acid, Glycoursodeoxycholic acid, Glycocholic acid
	Steroids		Pregnenolone sulfate, Androsterone glucuronide, Cholesterol 3-sulfate
	Ceramides (Cer)		HexCer-NS d34:2, SHexCer d34:1
	Sphingomyelins (SM)	SM 36:1, SM 36:2	SM 34:1
	Monoacylglycerols (MG)	MG 18:1, MG 18:2	
	Diacylglycerols (DG)	DG 36:2, DG 36:3, DG 36:4	
	Triacylglycerol (TG)	TG 54:6	
	Fatty acids	Oleic acid, Oleamide, Hydroperoxy-octadecadienoic acid, Dihydroxy-octadecanoic acid, Epoxyeicosatrienoic acid, N-Oleoyltaurine, Hydroxyhexadecanoic acid, 8-Hydroxylinoleic acid, Eicosatrienoic acid, Arachidonic acid, Linoleic acid	Dodecenoic acid
	Lysophosphatidylcholine (LPC)		LPC 14:0, LPC 15:0, LPC 16:0, LPC 16:1, LPC 17:0, LPC 18:0, LPC 18:1, LPC 18:2, LPC 20:0, LPC 20:1, LPC 20:4, LPC O-18:1
	Lysophosphatidylethanolamines (LPE)		LPE 16:0, LPE 18:0, LPE 18:1, LPE 18:2, LPE 20:4, LPE 22:6
	Phosphatidylcholines (PC)	PC 31:2, PC 31:3, PC 36:4, PC 38:4, PC 40:6	PC 30:0, PC 32:0, PC 32:1, PC 32:2, PC 33:0, PC 34:1, PC 34:2, PC 34:3, PC 35:2, PC 36:2, PC 38:5, PC 40:5, PC(P-36:4), PC(P-38:4), PC(P-38:5)
	Phosphatidylinositols (PI)	PI 32:1, PI 38:4	PI 36:2, PI 36:3, PI 38:5
	Phosphoethanolamines (PE)	PE 34:1, PE 34:2, PE 36:4, PE 38:4, PE 38:5, PE 38:6, PE 40:6	PE 34:3, PE 40:4, PE 40:5, PE(P 40:6)

Gly, glycine; Val, valine; Ile, isoleucine; Leu, leucine; Phe, phenylalanine; Trp, tryptophan; HexCer, hexosylceramide.

**Table 3 ijms-24-07761-t003:** Classes of metabolites (**A**) Only increased in serum samples of MO patients compared to normal weight subjects; (**B**) Only decreased in serum samples of MO patients compared to normal weight subjects.

(A)		
Group	Class	Increased Metabolite Levels
Amino acids, peptides and choline	Cholines	Choline
Lipids and derivatives	Monoacylglycerols (MG)	MG 18:1, MG 18:2
	Diacylglycerols (DG)	DG 36:2, DG 36:3, DG 36:4
	Triacylglycerol (TG)	TG 54:6
**(B)**		
**Group**	**Class**	**Decreased Metabolite Levels**
Lipids and derivatives	Bile acids	Deoxycholic acid, Glycoursodeoxycholic acid, Glycocholic acid
	Steroids	Pregnenolone sulfate, Androsterone glucuronide, Cholesterol 3-sulfate
	Ceramides (Cer)	HexCer-NS d34:2, SHexCer d34:1
	Lysophosphatidylcholine (LPC)	LPC 14:0, LPC 15:0, LPC 16:0, LPC 16:1, LPC 17:0, LPC 18:0, LPC 18:1, LPC 18:2, LPC 20:0, LPC 20:1, LPC 20:4, LPC O-18:1
	Lyso-phosphatidylethanolamines (LPE)	LPE 16:0, LPE 18:0, LPE 18:1, LPE 18:2, LPE 20:4, LPE 22:6

**Table 4 ijms-24-07761-t004:** Metabolite levels increased and decreased in serum samples of MO with T2DM patients compared to those metabolically healthy.

Group	Class	Increased Metabolite Levels	Decreased Metabolite Levels
Carbohydrates	Sugars	Anhydrohexose, Glucose, Sorbitol	Iditol
Amino acids, peptides and choline	Amino acids	Glutamate	
	Peptides		Phe-Ile/Leu
Lipids and derivatives	Acylcarnitines	Propionyl-carnitine	
	Bile acids	Glycoursodeoxycholic acid, Glycocholic acid	
	Ceramides (Cer)	HexCer-NS d34:2, SHexCer d34:1	
	Lysophosphatidylcholine (LPC)	LPC 14:0	
	Phosphatidylcholines (PC)	PC 30:0, PC 32:0, PC 32:2, PC 33:1, PC 36:0, PC 36:3, PC36:5	PC 32:1, PC 34:3
	Phosphatidylinositols (PI)	PI 36:3, PI 36:4	
	Phosphoethanolamines (PE)	PE 34:2, PE 34:3, PE 38:5, PE 38:6, PE 40:5, PE 40:6	

Ile, isoleucine; Leu, leucine; Phe, phenylalanine; HexCer, hexosylceramide.

## Data Availability

Data is unavailable due to privacy and ethical restrictions.

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
