# Peer review of "LC/MS-Based Untargeted Metabolomics Analysis in Women with Morbid Obesity and Associated Type 2 Diabetes Mellitus"

_ijms, 2023, doi:10.3390/ijms24097761_

Round 1

Reviewer 1 Report

The metabolomic analyzes have the bias of the methodology used to detect their markers. It is not clear from your study what it means to be an untargeted metabolomics study if the methodology used in its processing is liquid chromatography–mass spectrometry. On the other hand, it is clear that associating women with normal weight vs. morbidly obese women with or without diabetes or prediabetes in a single group explains why the PCA does not distinguish between the latter 2 groups. The group of obese women with prediabetes or diabetes should have been separated more precisely from obese women without glucose disorders. Even so, the findings found in the study coincide with other investigations, and the bias of doing so only in women could explain the new metabolites found. Finally, the order in which the paper is presented is not the usual one, as the methodology is generally presented before the results and then the discussion, conclusions and limitations of the study

Reviewer 2 Report

In this manuscript, Auguet et al. performed nontargeted metabolomics analysis in normal weight controls, morbid obese (MO) and morbid obese (MO) with type 2 diabetes. They identified a variety of metabolites that were different among these three groups.

Major points:

1.     In the introduction and abstract, the authors emphasized that it is important to determine the differential metabolites between metabolically healthy obese patients and those with metabolic disorders. However, they did not include healthy obese patients in their studies.

2.     In table 1, are there differences between MO patients and MO with T2DM? Can MO patients be considered as metabolically healthy obese?

3.     The authors listed metabolites that are increased or decreased between healthy control vs MO or MO vs MO with T2D. It is not clear what are the p values for these metabolites? From the results, it is difficult to appreciate how big the differences between groups are.

4.     The discussion section was not well written. Most of the discussion was just a repeat of results. 

5.     The authors should discuss more about how these metabolites, at least several most dramatically altered metabolites, may be involved in the obesity or T2D development.

Minor points:

1.     In table 1, the decimal point should be “.” not “,”

2.     On page 6, what does MHMO mean?

3.     On page 12, “2.4.2” and “2.4.3” should be 4.4.2 and 4.4.3. “2.5” should be 4.5.

Reviewer 3 Report

the study by Auguet  et al.,  conducted a metabolomics measurement in women with morbid obesity (MO) compared to a normal-weight group to examine the metabolome difference of these women with MO who present with type 2 diabetes. They performed an untargeted metabolomics measurement by liquid chromatography–mass spectrometry using serum samples of 209 Caucasian women: 73 with normal weight and 136 with MO, of which 65 had T2DM. First, they report elevated levels of choline and acylglycerols and reduced levels of bile acids, steroids, ceramides, glycosphingolipids, lyso-phosphatidylcholines and lyso-phosphatidylethanolamines in MO women than in the healthy control group. In MO women with T2DM, they report elevated levels of glutamate, propionil-carnitine, bile acids, ceramides, lyso-phosphatidylcholine 14:0, phosphatidylinositols and phosphoethanolamines and lower levels of Phe-Ile/Leu. Thus, they report metabolites with opposite trends of concentration in the two metabolomic analyses. They suggested that these metabolites could be considered as a potential novel  factors of study in the pathogenesis of MO and associated T2DM in women.

The manuscript is fine but I have few comments.

- the authors should add some limitations of the study such as small sample size and that the study did not study at the molecular levels the genes  that can be invoved in susceptibilty to morbid obesity, insulin resistance and T2D. It is well stablished that these disease are induced by interaction of genetic, epigenetic and enviroment. 

Author Response

First, we want to mention that we appreciate so much your contribution in this review process.

Then, regarding your comment on English grammar, to note that we sent the manuscript to an English grammar correction service with native staff (please see the attachment) although if you think it appropriate, we can ask for a second revision.

Finally, regarding your suggestion, we agree with your statement and we have added in the Discussion section the following sentence:

Moreover, this study has a small cohort of subjects and we did not evaluated at the molecular level the genes that can be involved in the susceptibility to develop MO, insulin resistance and T2DM, diseases induced by interaction of genetic, epigenetic and environmental factors [70-72]” (lines 348-351, page 11-12).

Reviewer 4 Report

In this report, the authors performed a comprehensive LC/MS-based untargeted metabolomics study in serum samples from 209 Caucasian women. The study cohort was composed of 73 healthy women with normal weight, who were categorized as the control group and 136 women with morbid obesity (65 were metabolically healthy and 71 with associated T2DM). The study classified the metabolites into three categories: carbohydrates, amino acids, peptides and choline and lipids and derivatives. The study revealed a clear differential metabolome analysis between subjects with T2DM associated with obesity analysis and morbid obesity subjects MO groups. Outcome study was quite different from previous metabolomics studies on obesity reporting differentially concentrated metabolites in different metabolic conditions which could be due to small size and heterogeneous cohorts that were performed on selected targeted metabolites.  Authors should elaborate more in linking their findings to cellular/biological pathways in order to explain differences in metabolites profiling between morbid obese with and without T2DM. Finding in this observational comparative study is of great interest, however, as stated by the authors the study needs to be validated on specific or targeted omics in larger.

Author Response

First of all, we would like to thank your collaboration in the revision of this article to be published in the IJMS.

Then, according to your suggestion asking for a more elaborate explanation about the linking of our results to cellular/biological pathways involved, we want to mention that in the updated version after the comments of reviewers 1 and 2, new information on the molecular/cellular role of the most relevant metabolites found in this study has already been added in the Discussion section (lines 334-345, page 11), however now we have tried to expand on this information:

Abnormal peptide and amino acid metabolism has been associated with impaired protein processing and accumulation of toxic intermediates that promote insulin resistance and consequently obesity [17,18]” (lines 194-196, page 9);

Concerning bile acids, which are involved in the intestinal absorption of fats and related to insulin resistance and obesity [44],[…]” (lines 240-241, page 10);

With regard to sphingomyelins, which have been related to insulin signalling inhibition and induction of inflammation [48],” (lines 249-250, page 10);

In relation to the complex phospholipids found in cell membranes [49], in this study we found […]” (lines 254-255, page 10)

In both obesity and diabetes, the families of metabolites found belong to the same metabolic pathways, although there are differences in the levels of some metabolites. For this reason, it is very difficult to explain the differences in the pathways between the diseases, and validation studies in larger cohorts are needed to confirm in which metabolic pathways these metabolites are involved in relation to obesity and type 2 diabetes mellitus.

Finally, concerning your last comment, we are aware that this study needs to be validated in other cohorts and hope that our group or others can do such validation soon.

Round 2

Reviewer 2 Report

The authors have addressed my concerns.

Author Response

Many thanks for your valuable input in the review process of this mansucript.